# Analytic Validation of Optical Genome Mapping in Hematological Malignancies

**DOI:** 10.3390/biomedicines11123263

**Published:** 2023-12-09

**Authors:** Andy W. C. Pang, Karena Kosco, Nikhil S. Sahajpal, Arthi Sridhar, Jen Hauenstein, Benjamin Clifford, Joey Estabrook, Alex D. Chitsazan, Trilochan Sahoo, Anwar Iqbal, Ravindra Kolhe, Gordana Raca, Alex R. Hastie, Alka Chaubey

**Affiliations:** 1Bionano, San Diego, CA 92121, USA; apang@bionano.com (A.W.C.P.);; 2Bionano Laboratories, San Diego, CA 92121, USA; 3Department of Pathology, Medical College of Georgia, Augusta University, Augusta, GA 30912, USA; 4DNA Microarray CGH Laboratory, Department of Pathology, University of Rochester Medical Center, Rochester, NY 14642, USA; 5Department of Pathology and Laboratory Medicine, Children’s Hospital of Los Angeles, Los Angeles, CA 90027, USA

**Keywords:** optical genome mapping (OGM), structural variation (SV), copy-number variation (CNV), standard of care (SOC)

## Abstract

Structural variations (SVs) play a key role in the pathogenicity of hematological malignancies. Standard-of-care (SOC) methods such as karyotyping and fluorescence in situ hybridization (FISH), which have been employed globally for the past three decades, have significant limitations in terms of resolution and the number of recurrent aberrations that can be simultaneously assessed, respectively. Next-generation sequencing (NGS)-based technologies are now widely used to detect clinically significant sequence variants but are limited in their ability to accurately detect SVs. Optical genome mapping (OGM) is an emerging technology enabling the genome-wide detection of all classes of SVs at a significantly higher resolution than karyotyping and FISH. OGM requires neither cultured cells nor amplification of DNA, addressing the limitations of culture and amplification biases. This study reports the clinical validation of OGM as a laboratory-developed test (LDT) according to stringent regulatory (CAP/CLIA) guidelines for genome-wide SV detection in different hematological malignancies. In total, 60 cases with hematological malignancies (of various subtypes), 18 controls, and 2 cancer cell lines were used for this study. Ultra-high-molecular-weight DNA was extracted from the samples, fluorescently labeled, and run on the Bionano Saphyr system. A total of 215 datasets, Inc.luding replicates, were generated, and analyzed successfully. Sample data were then analyzed using either disease-specific or pan-cancer-specific BED files to prioritize calls that are known to be diagnostically or prognostically relevant. Sensitivity, specificity, and reproducibility were 100%, 100%, and 96%, respectively. Following the validation, 14 cases and 10 controls were run and analyzed using OGM at three outside laboratories showing reproducibility of 96.4%. OGM found more clinically relevant SVs compared to SOC testing due to its ability to detect all classes of SVs at higher resolution. The results of this validation study demonstrate the superiority of OGM over traditional SOC methods for the detection of SVs for the accurate diagnosis of various hematological malignancies.

## 1. Introduction

Hematological malignancies refer to a distInc.t group of neoplastic diseases of hematopoietic and lymphoid tissues and are broadly divided into myeloproliferative neoplasms, myelodysplastic neoplasms, leukemias, lymphomas, and plasma cell neoplasms. Historically, these malignancies have been genetically characterized for diagnosis, classification, prognostication, and therapeutic decision making [1,2,3]. In the past decade, the genetic testing of these malignancies has dramatically evolved with the advancement of genomic technologies, particularly molecular characterization, due to the advent of next-generation sequencing (NGS) technology [4,5,6]. Although there have been significant improvements in molecular characterization of such blood tumors with NGS, the evolution of cytogenetic analysis has lagged comparatively, still relying on traditional methods, Inc.luding karyotyping (KT); fluorescence in situ hybridization (FISH); and, in some cases, chromosomal microarray (CMA).

Karyotyping, which is currently the “gold-standard” cytogenetic method for the detection of single-cell genome-wide structural variation (SV), suffers from several limitations: (1) low resolution of aberration sizes (10–20 Mbp), (2) cryptic translocations that remain undetected, and (3) a lack of metaphase cells in certain malignancies, resulting in failed KT (e.g., CD138+ cells in plasma cell myeloma and lymph-node single-cell suspension in lymphoma) [7,8]. Targeted FISH assays are often performed alongside KT or in isolation (for CD138+, lymph-node single-cell suspension or disease monitoring) to capture a limited number of SVs. CMA has seen limited adoption for hematological malignancies; despite its usefulness for copy-number variation (CNV) detection, it fails to detect insertions, balanced SVs (translocations and inversions), or fusions [9]. The aforementioned limitations of these cytogenetic/cytogenomic methodologies necessitate the use of multiple assays to obtain a reasonable cytogenetic profile in the majority of these cases [3,10,11]. Recently, NGS has been explored as a method to detect cytogenetic aberrations [12], but the intrinsic limitations involving repetitive sequences of the genome result in limited resolution and the inability to detect many SVs [13]. NGS performs well for the detection of small variants such as SNVs and indels of clinical relevance but requires multiplexing, complex bioinformatics, and high depth of coverage and is associated with high costs for utility in detection of cytogenetic aberrations. For instance, whole-genome sequencing was recently used to find cytogenetic aberrations in myeloid acute myeloid leukemia (AML) and myelodysplastic syndromes (MDS). The results were promising, but they only considered recurrent SVs and CNVs of at least 5 Mbp, excluding smaller CNVs and additional translocations; the results were also generated through extensive bioinformatic analysis [12]. Therefore, NGS is restricted to the sequencing of gene panels in hematologic malignancies as a complement to cytogenetic techniques in most labs.

Recently, multiple studies have demonstrated that optical genome mapping (OGM) is a powerful modern cytogenomic technology that provides a streamlined workflow at a high precision for the detection of all classes of SV at a genome-wide level [14,15,16,17]. These studies have shown ~100% concordance of OGM with classical cytogenetic methods (KT, FISH, and CMA) in most hematological malignancies [18,19,20,21,22,23,24]. In addition, OGM has demonstrated the ability to detect additional clinically relevant SVs missed by SOC, owing to its significantly higher sensitivity and resolution (~10,000× compared to KT), indicating that OGM is a viable alternative to traditional cytogenetic tests.

In this multisite, institutional review board (IRB)-approved study on hematological malignancies, a clinical validation of OGM was conducted for the development of a laboratory-developed test (LDT) in CLIA certified laboratories (Clinical Laboratory Improvement Amendments (CLIA) provide standards and have the authority to regulate certain clinical testing). The samples were evaluated for concordance, reproducibility, and assay robustness, and protocols were established for analysis and interpretation using guidelines-based targeted variant assessment, in addition to a whole-genome analysis. The unique ability of OGM to detect all classes of balanced and unbalanced SV at high resolution and Inc.reased sensitivity holds promise for its acceptance as a first-tier cytogenomic test for all hematological malignancies. This is in line with multiple published studies demonstrating the reproducibility and robustness of the OGM workflow, as it simplifies the operations of the clinical testing laboratory.

## 2. Materials and Methods

### 2.1. Cohort Design

Multiple U.S.-based laboratories contributed to this double-blinded observational study for sample recruitment, data collection, and variant analysis. The study protocol was approved through multiple institutional review boards (IRBs) and Inc.luded consent provided by individuals with newly collected samples or waived authorization for use of deidentified samples. All protected health information (PHI) was removed, and data were anonymized (coded and double-blinded) before accessioning for the study. Samples were given anonymous aliases used in this study (e.g., BNGOHM-xxxxxxx). All clinical samples (peripheral blood or bone marrow aspirate) were referred for clinical cytogenetic testing due to a suspicion of a hematological malignancy, and SOC test results Inc.luding KT, FISH, and/or CMA were available. For the deidentified cases, clinical indications, genetic test results, and additional demographic information were collected as available. Control specimens Inc.luded either established cell lines, peripheral blood, or bone marrow samples from healthy adults.

### 2.2. OGM Assay Workflow (DNA Isolation, DNA Labeling, Chip Loading, and Data Collection)

Frozen aliquots of cells, bone marrow aspirate (BMA), and/or peripheral blood (PB) were subjected to DNA isolation using manufacturer protocol of the OGM DNA Isolation Procedure, v3.0 (Bionano Genomics Inc., San Diego, CA, USA). Briefly, frozen sample vials were thawed in a 37 °C water bath; then, counted for number of cells was using a HemoCue WBC Analyzer (Fisher Scientific, Waltham, MA, USA). A total of 1.5 million cells per sample were transferred to Protein Lo-Bind microfuge tubes (Eppendorf, Enfield, CT, USA) for centrifugation. Cell pellets were resuspended and washed with stabilizing buffer. Washed cell suspensions were enzymatically digested and lysed, and isopropanol was used to precipitate ultra-high-molecular-weight (UHMW) DNA in the presence of a nanobind disk. Long DNA strands bound to nanodisks were washed, transferred to clean tubes, and subsequently released from the nanodisk using elution buffer.

Following the procedure outlined in Bionano Prep Direct Label and Stain (DLS) protocol (Bionano Genomics Inc., San Diego, CA, USA), approximately 500 ng–750 ng of solubilized UHMW DNA was labeled enzymatically, conjugating fluorophores to the target 6-base sequence motif, *CTTAAG*. Long, labeled DNA strands were then counterstained with an intercalating dye, homogenized in buffer to allow for flow through a nanochannel device, and loaded into the flow cells of Saphyr G2.3 chips (Bionano Genomics Inc., San Diego, CA, USA). The chips were run on a Saphyr instrument (Bionano Genomics Inc., San Diego, CA, USA) to a target throughput >1500 Gbp per sample (OGM Saphyr Chip Loading and Data Collection Procedure, v3.0).

### 2.3. Assay QC

The completed datasets were then assessed for the following analytical quality control metrics: 320X effective coverage of GRCh38, with ≥70% of molecules ≥ 150 kbp aligning (“map rate”) at an N50 of ≥230 kbp. Additionally, the Bionano Access 1.7 EnFocus^TM^ (Fragile X) pipeline was run for a subset of samples chosen randomly to assess post-analytical QC pass/fail metrics (CNV noise and stable region analysis) and to infer the gender for each case. N50 is defined as the length of the shortest molecule for which the sum of the lengths of the longer molecules is greater than 50% of the total length of all molecules.

### 2.4. SV Detection Using Rare Variant Pipeline

The Bionano Solve (version 3.7) rare variant pipeline was used for genome-wide SV detection. The rare variant pipeline enables the detection of SVs occurring at low allelic fractions. Molecules were aligned to the GRCh38 reference, and clusters of molecules (≥3) indicating SVs were used for local assembly. Local consensus assemblies have high accuracy and are used to make final SV calls by realignment to the reference genome. SV calls were finally compared against known genes and against SVs in an OGM control SV database with 179 population controls.

Separately, an analysis based on relative read depth, a copy-number profile that can identify gains and losses, was used. Putative copy changes were segmented, and calls were generated and similarly annotated with positional information from the original reference. Entire chromosomal aneusomies were likewise defined in the CNV algorithm.

### 2.5. Post-Analytical SV Curation and Classification

An SV filtering and curation protocol was devised and implemented using Bionano Access (version 1.7.2). An overview of the subsequent curation and classification procedure is shown in Figure 1. For the concordance part of the study, a set of filters was applied to Inc.lude variants with a variant allele fraction (VAF) of 0.02–1 and present in 0% of controls (built in the control SV database). SVs meeting these criteria were then curated for manual review and classification. The analyst remained blinded until the classification was performed. SVs were classified using a tiering system based on the ACMG/CGC guidelines adopted for OGM. Briefly, the four-tiered evidence-based SV classification followed professional recommendations for interpretation, variant classification, and reporting of genomic findings in neoplastic disorders [25], which were, in turn, informed by guidelines resourced form the World Health Organization (WHO), the National Comprehensive Cancer Network (NCCN), and the National Health Service (NHS, UK). For all samples used in the concordance, specificity, intersite replication parts of the study, curation and classification were performed in four successively applied filtering steps: disease subtype-specific classification (first, when applicable; most stringent filter), pan-hematological malignancy classification, pan-cancer classification, and remaining variant classification (last and most permissive filter) (Figure 1).

Following validation, cases were run and analyzed according to a standardized procedure in an end-to-end exercise. In these cases, after variant analysis and preliminary classification, draft reports were reviewed by board-certified pathologists or laboratory directors for final classification, interpretation, and determination of readiness for future reporting. Directors proceeded through classified variants in the curated list, upholding and/or revising analyst classification as needed. Upon completing their reviews of the curated/classified variant list, the directors finalized case summary statements regarding somatic variants and defined the genomic complexity status (defined as normal if no large (≥5 Mbp) aberrations were detected, simple if fewer than three were detected, and complex for cases containing at least three aberrations for diseases other than acute lymphocytic leukemia (ALL) and for ALL cases containing at least five aberrations).

### 2.6. Concordance with SOC

Concordance analysis was performed between SVs detected using SOC testing and OGM data. Most of the samples had KT or KT plus FISH SOC results, and a subset had CMA data. If the same SV was observed by more than one SOC method, it was considered a single SV for concordance purposes. An SV was scored as concordant if the chromosome and band matched. Any difference in size of break points was attributed to technique differences and the higher resolution of OGM. Additionally, if multiple OGM calls supported one SOC variant, it was treated as a single concordant event. In addition to the concordance assessment, each case was evaluated for additional pathogenic/likely pathogenic (tier 1 and 2, respectively) findings that were not detected by SOC. Orthogonal methods were used to confirm a subset of these additional SVs detected by OGM.

### 2.7. Intersite Replication

Concordance between replicates generated in different sites was performed on Tier 1 and 2 variants for cases and on the lack of Tier 1 and 2 variants for controls.

### 2.8. Reproducibility

UHMW DNA from four hematological malignancy BMA samples was labeled in different batches by different operators using multiple reagent lots and instruments. Reproducibility for each individual SV was calculated as the number of replicates where the SV was detected divided by the total number of replicates. Reproducibility for each SV type was calculated by considering the mean value across all variants for each type.

### 2.9. Limit of Detection

The limit of detection (LOD) was determined using two acute myelogenous leukemia (AML) cell lines from the ATCC: KG-1 and MV4-11 (labelled as BNGOHM-0000319 and BNGOHM-0000315, respectively, in this study). The DNA of each cell line was blended with GM12878 (normal control cell line, Coriell Institute) to create serial dilutions that ranged from undiluted to 1:24 dilution. Combined samples were gently mixed over the course of one week to ensure uniform mixing of DNA molecules. Six replicates of each blend were run through OGM with a target of >1500 Gbp of DNA and analyzed with the rare variant pipeline.

## 3. Results

In this study, samples from 60 cases with hematological malignancies (with various heme subtypes), 2 cancer cell lines, and 18 controls were used (*n* = 80, Figure 2). From these clinical cases, cell lines, and controls, 215 total datasets were generated. The molecular N50, map rate, and effective coverage were recorded as analytical quality metrics, averaging 263 kbp, 87%, and 434×, respectively (Appendix A).

### 3.1. Concordance with SOC and Inc.reased Yield

Concordance with SOC testing results was conducted based on 77 datapoints, and OGM demonstrated 100% concordance. Detailed concordance results can be found in Appendix A.

OGM was assessed for its ability to identify known gains and losses impacting genes associated with hematologic neoplasms. In the complex genome of a case (BNGOHM-0000191) with myelodysplastic syndromes (MDS), OGM detected a 392.8 kbp tandem duplication on chr7, resulting in the duplication of *KMT2C, GALNTT11*, and *GALNT5* (Figure 3a). Inc.reased *GALNT11* expression has been reported for leukemia, and *KMT2C* has been associated with tumorigenesis. Furthermore, both the map-based OGM SV algorithm and the coverage-based CNV algorithm detected an 18.8 Mbp heterozygous deletion on chr20 in a myeloproliferative neoplasms (MPN) genome (BNGOHM-0000176), impacting multiple genes associated with myeloid neoplasms, such as *L3MBTL1* and *SGK2* (Figure 3b). In an ALL case (BNGOHM-0000172), a gain of chromosome Y was detected with the depth-of-coverage aneuploidy algorithm (Figure 3c). Although a gain of Y may be associated with constitutional disorders, OGM was able to confirm the sex chromosome aneuploidy detected by SOC.

In addition, OGM achieved 100% concordance in more complicated aberrations, such as translocations and inversions (Appendix A). For instance, in the BNGOHM-0000138 sample (MDS), an unbalanced t(2;7) translocation was called where the map captured the translocation breakpoint, while the depth-of-coverage profiles indicated a gain in 2p and a loss in 7q (Figure 3d). Also, in MDS sample BNGOHM-0000191, OGM called a t(2;15) followed by an 168.9 kbp inversion, whose breakpoint interrupted *BUB1B*, a kinase gene involved in spindle checkpoint function (Figure 3e). The ability to resolve the local structure of these two neighboring SVs was made possible by OGM’s ultra-long molecules.

Moreover, OGM resolved structures in complex genomes. For example, OGM, in combination with KT and FISH, confirmed that acute lymphocytic leukemia (ALL) case BNGOHM-0000333 carried multiple whole-chromosome duplications (4, 6, 8, 10, 12, 14, 16, 17, 18, and X), as well as a triplication of chromosome 21 (Figure 3f). These events indicate hyperdiploidy, which has been observed in ALL genomes. Also, OGM resolved a complex case of ALL (sample = BNGOHM-0000243) with higher resolution than KT and FISH. The loss of chr9 could be the result of a series of DNA segments translocated into other regions on chromosomes 4, 9, 12, and 21 via inter- and intrachromosomal fusions. In addition, CMA had previously captured multiple interstitial deletions. OGM not only recalled the deletions, but it also found translocations at the break points (Figure 3g). Therefore, these DNA losses were the results of unbalanced translocations. Therefore, OGM effectively consolidated the results of three orthogonal technologies.

Finally, in addition to confirmation by SOC, OGM identified additional novel Tier 1 and 2 variants not previously reported by SOC in 17 out of 60 (28%) cases; a subset of these variants was subsequently confirmed by orthogonal methods (Appendix A). In one case of a myeloid neoplasm, BNGOHM-0000149, OGM was able to uniquely identify a mosaic deletion of 3q13.31 to 3q22.3 overlapping *GATA2,* a gene associated with myelogenous leukemia and an Inc.lusion criterion for at least one clinical trial (NCT01861106), as well as complex rearrangements on 17p1, consisting of amplifications and deletions that impacted *YWHAE*, *MNT*, *TP53*, and *MAP2K4* (Figure 4a). In a second case with biphenotypic leukemia (AML/ALL) indicated at the time of collection, i.e., BNGOHM-0000335, OGM detected a translocation between chromosomes 16 and 12 (t(12;16)(p13;p13)) not reported by SOC. This translocation resulted in the *CREBBP-ZNF384* fusion gene, which is an abnormality reported in ALL (Figure 4b).

### 3.2. Intersite Replication

There were 53 datapoints used to assess the intersite reproducibility of the assay and workflow. Among the 53 datapoints, 22 were controls with no reported rearrangements across sites. Among the remaining cancer datapoints, we evaluated the reproducibility of 56 Tier 1 or 2 variants: 54 (96.4%) were concordant, 1 (1.8%) was partially concordant, and 1 (1.8%) was discordant (Appendix A). The one partially concordant locus and one discordant locus both originated from one case, where the replicate had suboptimal copy-number quality metrics. For the partially concordant locus, the poorer quality replicate only called a low-level partial 16q arm loss, whereas the initial test called a whole arm loss. As for the discordant locus, the replicate test failed to identify a very low-level loss of 20p.

### 3.3. Specificity

All Inc.idental SVs that were detected in 18 healthy donor blood samples were compared to both a list of 206 targets/types recommended for testing by medical associations (NCCN, WHO, and NHS (Appendix A)) and a comprehensive list of cancer genes. Overall, Tier 1 or 2 heme-associated variants were only found in two of the healthy donor samples. A loss of Y was detected in BNGOHM-0000076, and a 71 kbp deletion detected in 9p24.1 impacting the *PDCD1LG2* gene was identified in another donor (anonymized ID: 1000107245). Both variants were subsequently confirmed by CMA as true-positive calls. In summary, based on this analysis, there were no false-positive SVs detected in the genomes of healthy donors, indicating 100% specificity (for Tier 1 and 2 SVs).

### 3.4. Reproducibility

Reproducibility for each SV was determined by calculating the fraction of replicates (N ≥ 6) in which each variant was accurately identified (Appendix A). Overall, the reproducibility assay Inc.luded an aggregate of 227 SVs (14 aneusomies, 46 duplications, 33 insertions, 58 deletions, 12 inversions, 32 intrachromosomal fusions, and 32 interchromosomal translocations). In total, there was 96% reproducibility among replicates for all SVs, CNVs, and aneuploidies. A total of five variants were not detected across all replicates: two duplications, two insertions, and one inversion. The lack of detection was attributed to one of the following: SV residing in regions with segmental duplications, SV having low variant confidence score, or SV residing in high complexity regions.

### 3.5. Limit of Detection

Two cell lines (BNGOHM-0000319 and BNGOHM-0000315) were used to evaluate the limit of detection (LOD) of OGM. Two deletions, three translocations, an inversion, and two trisomies were assessed in a dilution series with six replicates at each dilution level. In summary, at the current coverage of >1500 Gbp, map-based SV calling can detect deletions at ≥5% VAF, inversions at ≥5% VAF, translocations at ≥5% VAF, and trisomies at a fractional copy number of 2.25 (Figure 5).

## 4. Discussion

Medical guidelines provided by professional societies such as the NCCN, WHO, NHS, etc., recommend testing for SVs as part of the workup of suspected hematologic malignancies to establish diagnosis, monitor disease progression, and direct management. Current laboratory practices rely on several complementary cytogenetic techniques to accomplish recommended testing, but these methods suffer from the limitation of low resolution, therefore missing actionable SVs in some cases. Cytogenetic/cytogenomic testing laboratories often have customized testing algorithms depending on the preliminary diagnosis and disease stage. Physician preferences and institutional testing and reimbursement policies often determine the types and extent of diagnostic testing. These decades-long disparities typically lead to different combinations of FISH and/or KT testing depending on the laboratory infrastructure, instrumentation, and availability of trained cytogenetics laboratory professionals. There is a need for innovative technologies that can overcome the challenges of traditional SOC testing and introduce a comprehensive, uniform combination of tests being performed on a routine basis. This study demonstrates that OGM can be easily implemented in the clinical setting and can substantially reduce operational complexity and improve the detection rate by providing a reproducible and robust alternative to the three predominant cytogenetic methods (KT, FISH, and CMA) in routine workup of most hematological malignancies.

A wide variety of hematologic malignancies and SVs detected by multiple SOC methods were Inc.luded during this LDT validation process, and OGM detected 100% of these variants. OGM was validated for different SV classes, such as deletions, inversions, and translocations, all performing at or above the 5% VAF threshold. Importantly, the detection of novel, actionable, clinically significant SVs solely by OGM in a significant fraction of cases (17 out of 60 cases = 28%) highlights the substantial diagnostic benefits of OGM testing. One important feature of a robust assay is reproducibility; OGM detected over 96% of variants across all replicates within and between runs, as well as amongst test sites. 

The OGM technique involves the bulk analysis of the somatic variants, and parsing out subclonal information may be more difficult to ascertain compared with KT. However, OGM not only has very high sensitivity as compared to KT but also provides the variant allele fraction (VAF) of each variant to better understand the various abnormalities in these somatic samples. Although KT is a method that can consider individual clones, it fails to reveal cryptic translocation and is unable to ascertain the identity of marker chromosomes or additions on terminal arms of chromosomes and cannot define the break points involving oncogenes in complex abnormalities. OGM is unable to detect balanced centromeric translocations; however, professional societies such as the WHO and NCCN do not recommend such translocations for assessment in hematological malignancies. Our findings are in line with previous publications recommending OGM as a first-tier test for heme malignancies. In cases of complex genomes detected by OGM, KT might serve as a potential reflex test to assess disease-causing subclonal variants for a more comprehensive assessment of the cancer genome.

Another unique feature of the OGM assay involves the uniform preanalytical and analytical steps, irrespective of the disease subtype, which allow for workflow standardization and scalability for single or multiple hematologic malignancy subtypes. The Inc.orporation of guideline-driven, disease subtype-specific target variants into the analytical process allows for semiautomated and easy classification and reporting of variants. Additionally, if there is ambiguity in the clinical symptoms of a heme malignancy subtype at the time of the physician consult, a “pan-heme” genomic analysis can still be conducted simultaneously in a disease-agnostic fashion.

Multiple studies, Inc.luding the present study, demonstrate that OGM overcomes multiple limitations of SOC testing [11,14]. In agreement with recently published studies, this study demonstrates Inc.reased detection of clinically relevant SVs by OGM compared to SOC in 28% of cases (Tier 1 or Tier 2 SVs that were missed by SOC). Additionally, OGM provides a standardized data acquisition and analysis process with a software solution that allows for seamless and systematic implementation and adoption across multiple laboratories.

This clinical validation is the second published study in the USA according to CAP/CLIA guidelines [22]. After the validation was completed, a representative report template was created with Tier 1A variants highlighted on page 1 and Tier 1b/2 variants listed on page 2 (Figure 6). Section A is analogous to a FISH panel, with present/absent indicated for each abnormality. Section B represents a whole-genome analysis and displays the SVs detected on a chromosome-by-chromosome basis (analogous to a KT).

## 5. Conclusions

OGM is a viable alternative to multiple methods, such as, KT, FISH, and CMA, addressing several lacunae of the traditional methods. The OGM workflow provides an end-to-end solution, from DNA isolation to downstream SV analysis and interpretation, using analysis software to facilitate adoption in a clinical laboratory. SInc.e the OGM assay does not require culturing of the clinical specimens, the typically observed culture biases in a cytogenetic laboratory are eliminated. OGM also allows for a sample to answer to be obtained in 4–5 days, which is extremely beneficial for the management and treatment of these patients. OGM not only detects all classes of SVs but provides the highest resolution attained to date by any cytogenetic method in clinical use. The ability of OGM to detect both recurrent SVs and novel fusions positions it as a first-tier test for the detection of all classes of SVs in potentially all hematological malignancies.

## Figures and Tables

**Figure 1 biomedicines-11-03263-f001:**
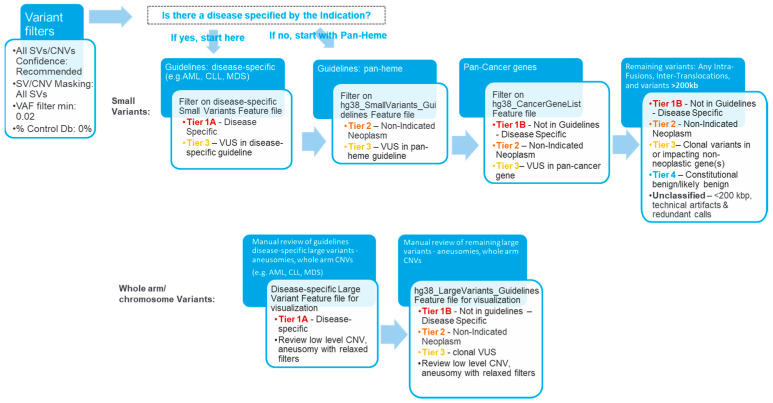
Overview of the analysis and interpretation workflow. The curation and classification process consists of four steps: (1) Disease-specific: Several organizations, such as the World Health Organization (WHO), the National Health Service (NHS), and the National Comprehensive Cancer Network (NCCN), have published guidelines for assessing clinically relevant genomic regions for hematological malignancies. The analysts apply a filter to select for variants overlapping with known loci associated with the disease of the sample (e.g., AML), then classify the selected SVs as Tier 1A or 3. (2) Pan-hematological cancers: The analyst removes the first filter and applies another filter to select for SVs seen in hematological cancers and classifies them as Tier 2 or 3. (3) Pan-cancer: SVs overlapping other cancer-associated genes are selected and classified as Tier 1B, 2, 3, or 4. (4) Remaining variants: All remaining fusion SVs or SVs >200 kbp are classified as Tier 1B, 2, 3, or 4.

**Figure 2 biomedicines-11-03263-f002:**
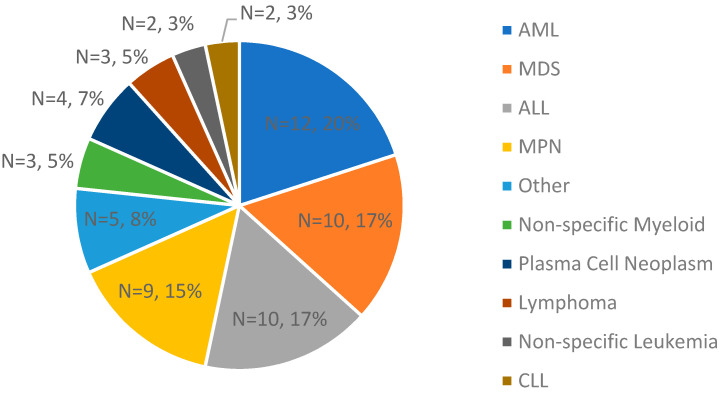
A breakdown of clinical indications of the cases used in validation for concordance assessment (*n* = 60; the two cell lines are not Inc.luded).

**Figure 3 biomedicines-11-03263-f003:**
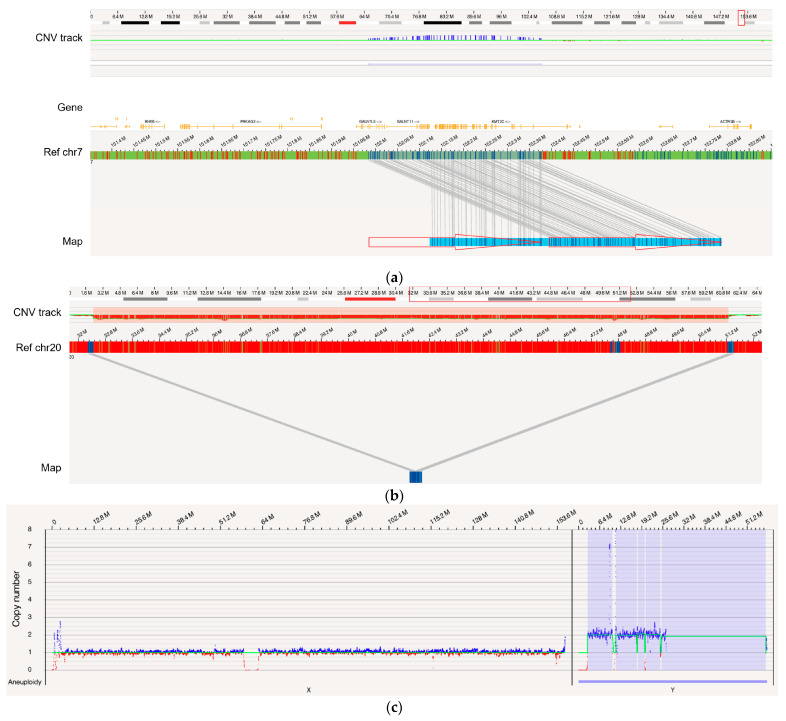
Examples of several types of SVs examined in comparison with SOC. (**a**) A 392.8 kbp tandem duplication occurred in the genome of BNGOHM-0000191, an MDS sample, with a VAF of 1.0. The two red arrows indicate the two copies of the duplicon on the assembled map. (**b**) A 18.8 Mbp deletion with a VAF of 0.44 was observed in MPN genome BNGOHM-0000176. Note that both the coverage-based copy-number algorithm and the map-alignment-based algorithm captured the deletion. (**c**) In ALL sample BNGOHM-0000172, a whole-genome duplication of chromosome Y was observed. (**d**) OGM captured an unbalanced t(2;7) in the MDS genome of sample BNGOHM-0000138. The blue and red lines indicate the translocation break points, which coInc.ide with a copy-number gain on chr2 and a loss on chr7. The VAF of the translocation was 0.28. (**e**) A 168.9 kbp inversion adjacent to a t(2;15) translocation was captured by a long genome map. The events occurred at a VAF of 0.27, and the *BUB1B* gene was interrupted by the inversion break point. (**f**) A hyperdiploid genome in ALL genome BNGOHM-0000333 as seen in OGM’s whole-genome coverage view. (**g**) OGM resolved the structure of a complex ALL genome of sample BNGOHM-0000243.

**Figure 4 biomedicines-11-03263-f004:**
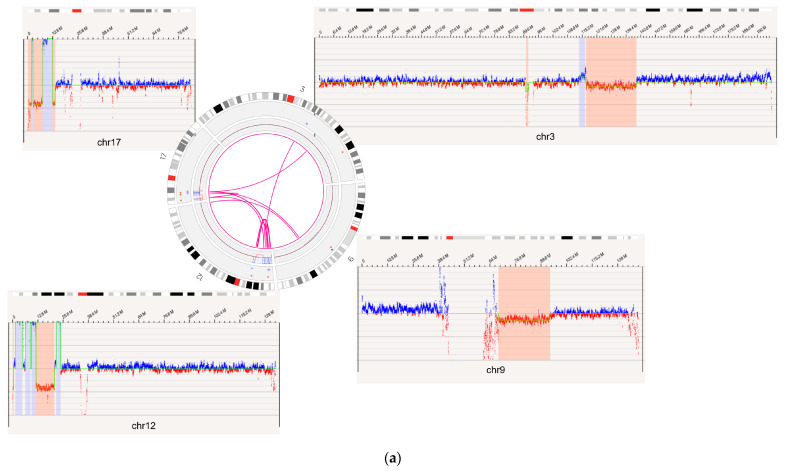
OGM captured additional novel Tier 1 and 2 findings not detected by SOC methods. (**a**) Complex rearrangements that consist of a deletion of the *GATA2* gene on 3q13.31 to 3q22.3 and multiple variants on 17p13 seen in the case of BNGOHM-0000149. The Circos plot and coverage profiles of chromosomes 3, 9, 12, and 17 are shown, all of which are associated with the 17p rearrangements. (**b**) The *CREBBP-ZNF384* fusion (**top**) and its reciprocal translocation (**bottom**) captured by OGM in the case of BNGOHM-0000335. The first picture shows the conjoining of chr16 with chr12 such that *CREBBP* would be fused with *ZNF384*, and the second shows the reciprocal fusion point.

**Figure 5 biomedicines-11-03263-f005:**
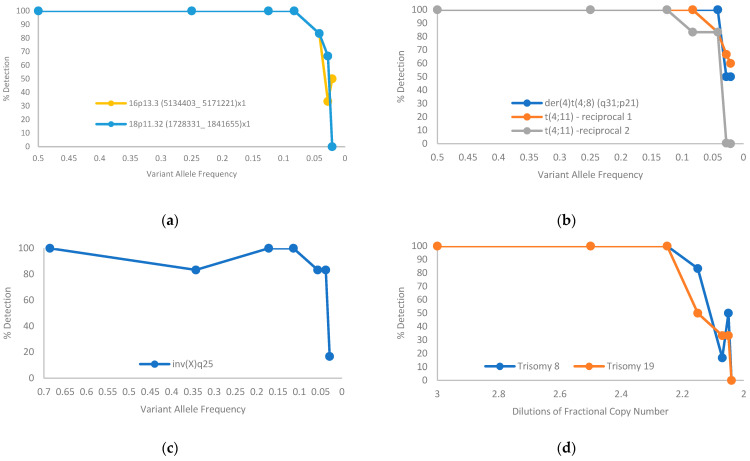
Limit-of-detection of aberrations of SVs estimated from serial dilution of two cancer cell lines: (**a**) deletion; (**b**) translocation; (**c**) inversion; (**d**) trisomy.

**Figure 6 biomedicines-11-03263-f006:**
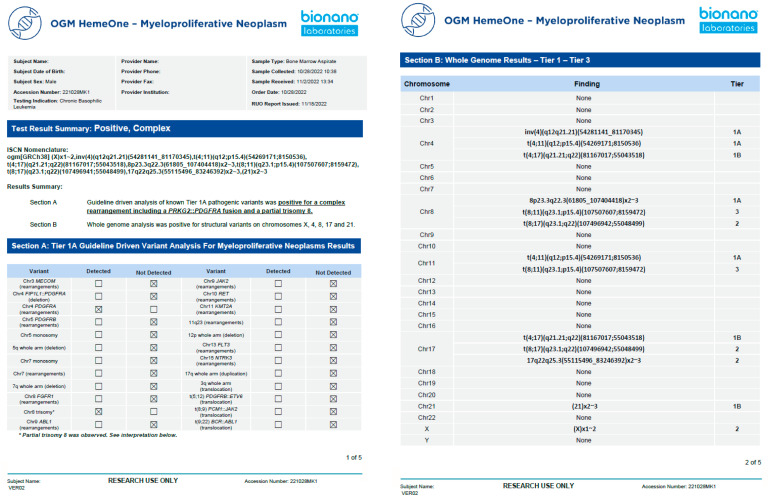
Example of a representative clinical OGM report. After demographic information, a summary of test results reports whether any variant was detected (positive/negative) and if the genome was simple or complex. Section A shows the defined Tier 1A abnormalities as a panel, and section B shows the whole-genome profile from chromosomes 1-22, X, and Y.

## Data Availability

Data will be made available upon reasonable request and in accordance with IRB protocols.

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
