# Peer review of "Analytic Validation of Optical Genome Mapping in Hematological Malignancies"

_biomedicines, 2023, doi:10.3390/biomedicines11123263_

Round 1
Reviewer 1 Report
Comments and Suggestions for Authors
In this validation study, Pang et al. tested 68 cases of various subtypes of hematological malignancies, 27 controls and two cancer cell lines with the Bionano Saphyr system to perform Optical genome mapping (OGM) analysis for genome-wide structural variant (SV) detection and to assess its feasibility in clinical settings as a laboratory developed test (LDT), according to CLIA guidelines. They generated 215 datasets and analyzed the data using files containing lists of diagnostically or prognostically relevant genomic regions obtained from guidelines of World Health Organization (WHO), the National Health Service (NHS) and the National Comprehensive Cancer Network (NCCN) with which they prioritize calls as disease-specific or pan-cancer specific. They established that sensitivity, specificity, and reproducibility of the method were 100%, 100% and 96%, respectively. Indeed, as reported in several previous studies, they showed that OGM found more clinically relevant SVs compared to standard-of-care (SOC) methods due to its ability to detect all classes of SVs at higher resolution. Concluding, they propose that the highest performance of OGM to detect both recurrent SVs and novel fusions could render it as a first-tier test to be performed in clinical settings “for the detection of all classes of SVs in majority of the hematological malignancies” and suggest a representative report template to show analysis’s results.
This report is the second clinical validation performed in USA according to CAP/CLIA guidelines. This study is well designed and written and responds to the requests of clinicians and scientists to deepen cytogenetic analysis of hematological malignancies over the well-known limits of SOC methods and to perform a more comprehensive study of the analyzed patient. As considered the need to implement this technologic improvement in clinical settings in my opinion this study deserves publication in “Biomedicine”.
However, there are minor comments that I list below.
- It should be briefly described what the CAP/CLIA guidelines are (for non-American readers). What is IRB?
- Line 81, reference 15: this review describes only studies on AML, however in literature there are other collections describing studies on other hematologic malignancies. It would be appropriate to mention them (Chronic Lymphocytic Leukemia: Current Knowledge and Future Advances in Cytogenomic Testing. Wainman et al; Feasibility of Optical Genome Mapping in Cytogenetic Diagnostics of Hematological Neoplasms: A New Way to Look at DNA. Coccaro et al.)
- Throughout the manuscript you should check the acronyms (line 166, SOP)
- Lines 174-175: BNGOHM-0000335 was an acute lymphocytic or a myeloid leukemia case?
- Line 396: why did you talk of majority of hematological malignancies? What subtypes should not be evaluated with OGM?
- In Discussion part you listed the few limitations of OGM, however in Conclusions you propose it as a fist-ties test for analysis of hematological malignancies. Should it be accompanied with standard methods as karyotyping, as considered its limit in detecting some types of SVs or subclones? Or do you think that it should be conducted alone? You should express your opinion on this aspect.
- Figure 5: resolution must be improved, please.
- Figure S1: it should be moved to the manuscript as it describes the workflow of variants’ prioritization as disease-specific or pan-cancer specific. It should be improved to facilitate reader’s comprehension (for example the background should be white).
- Lines 159-160: “SVs were classified using a tiering system based on the ACMG guidelines adopted for OGM.” You should better specify how you generated the lists of variants (you cited also WHO, CNNC and NHS). Indeed, In the text you should better explain variants’ classification (Tier 1A, 1B, 2, 3, 4). In the Supplementary material you should list all the variants included in each classification (with the exception of Tier 4).
Author Response
Reviewer 1
Open Review
Comments and Suggestions for Authors
In this validation study, Pang et al. tested 68 cases of various subtypes of hematological malignancies, 27 controls and two cancer cell lines with the Bionano Saphyr system to perform Optical genome mapping (OGM) analysis for genome-wide structural variant (SV) detection and to assess its feasibility in clinical settings as a laboratory developed test (LDT), according to CLIA guidelines. They generated 215 datasets and analyzed the data using files containing lists of diagnostically or prognostically relevant genomic regions obtained from guidelines of World Health Organization (WHO), the National Health Service (NHS) and the National Comprehensive Cancer Network (NCCN) with which they prioritize calls as disease-specific or pan-cancer specific. They established that sensitivity, specificity, and reproducibility of the method were 100%, 100% and 96%, respectively. Indeed, as reported in several previous studies, they showed that OGM found more clinically relevant SVs compared to standard-of-care (SOC) methods due to its ability to detect all classes of SVs at higher resolution. Concluding, they propose that the highest performance of OGM to detect both recurrent SVs and novel fusions could render it as a first-tier test to be performed in clinical settings “for the detection of all classes of SVs in majority of the hematological malignancies” and suggest a representative report template to show analysis’s results.
This report is the second clinical validation performed in USA according to CAP/CLIA guidelines. This study is well designed and written and responds to the requests of clinicians and scientists to deepen cytogenetic analysis of hematological malignancies over the well-known limits of SOC methods and to perform a more comprehensive study of the analyzed patient. As considered the need to implement this technologic improvement in clinical settings in my opinion this study deserves publication in “Biomedicine”.
However, there are minor comments that I list below.
- It should be briefly described what the CAP/CLIA guidelines are (for non-American readers). What is IRB?
Thank you for the comments, we have now clarified in the manuscript.
The reference to CLIA/CAP guidelines in the introductory statements (Abstract, lines 26-27) refer to the requirements for an LDT to have undergone a rigorous pre-launch validation process to ensure that results will be accurate and exceed standards required for patient care and safety. In the US, CLIA and CAP accreditation guarantees laboratories providing high-complexity testing demonstrate compliance with professionally and scientifically sound and approved laboratory operating standards. Clinical test validation is an important requirement for these professional accreditation processes.
An IRB (Institutional Review Board) is an appropriately constituted peer group of professionals that have been formally designated by any institution to approve, oversee and monitor biomedical research involving humans or human samples. The IRB group reviews research protocols and processes to ensure patient patient’s rights (and data related to them) are appropriately protected.
We have modified the abstract to state: “This study reports the clinical validation of OGM as a laboratory developed test (LDT), according to stringent regulatory guidelines” (removing CAP/CLIA here)
We have modifed the 4th paragraph in the introduction to state: “In this multi-site, institutional review board (IRB)-approved study on hematological malignancies, a clinical validation of OGM was conducted for the development of a laboratory developed test (LDT) in CLIA certified laboratories (Clinical Laboratory Improvement Amendments (CLIA) provides standards and authority to regulate certain clinical testing).”
- Line 81, reference 15: this review describes only studies on AML, however in literature there are other collections describing studies on other hematologic malignancies. It would be appropriate to mention them (Chronic Lymphocytic Leukemia: Current Knowledge and Future Advances in Cytogenomic Testing. Wainman et al; Feasibility of Optical Genome Mapping in Cytogenetic Diagnostics of Hematological Neoplasms: A New Way to Look at DNA. Coccaro et al.)
We appreciate the good suggestion. We have added additional references:
Wainman, L. M.; Khan, W. A.; & Kaur, P. Chronic Lymphocytic Leukemia: Current Knowledge and Future Advances in Cytogenomic Testing. In C. M. Sergi (Ed.), Advancements in Cancer Research. Exon Publications, 2023.doi:10.36255/chronic-lymphocytic-leukemia.
Coccaro, N.; Anelli, L.; Zagaria, A.; Tarantini, F.; Cumbo, C.; Tota, G.; Minervini, C. F.; Minervini, A.; Conserva, M. R.; Redavid, I.; et al. Feasibility of Optical Genome Mapping in Cytogenetic Diagnostics of Hematological Neoplasms: A New Way to Look at DNA. Diagnostics (Basel, Switzerland), 2023, 13(11), 1841, doi.org/10.3390/diagnostics13111841.
Giguère, A.; Raymond-Bouchard, I.; Collin, V.; Claveau, J. S.; Hébert, J.; & LeBlanc, R. Optical Genome Mapping Reveals the Complex Genetic Landscape of Myeloma. Cancers, 2023, 15(19), 4687. doi.org/10.3390/cancers15194687
- Throughout the manuscript you should check the acronyms (line 166, SOP)
We thank the reviewer for this point. Acronyms have been checked for clarity at first use, and corrections made accordingly.
- Lines 174-175: BNGOHM-0000335 was an acute lymphocytic or a myeloid leukemia case?
The indication with the sample was “anemia and thrombocytopenia with blasts- our flow biphenotypic leukemia, AML/ALL”. We clarified the in-line text to “In a second case with biphenotypic leukemia (AML/ALL) indicated at time of collection, …”
- Line 396: why did you talk of majority of hematological malignancies? What subtypes should not be evaluated with OGM?
This is a good point, we intended to indicate that publications exist the survey most hematologic malignancies, but there are some subclasses that have not been assessed sufficiently to extrapolate to all heme malignancies subclasses. We have adjusted to leave it open to most or all: “The ability of OGM to detect all classes of SVs including recurrent CNVs, SVs, and novel fusions positions it as a potential first-tier test for all hematological malignancies.”
- In Discussion part you listed the few limitations of OGM, however in Conclusions you propose it as a fist-ties test for analysis of hematological malignancies. Should it be accompanied with standard methods as karyotyping, as considered its limit in detecting some types of SVs or subclones? Or do you think that it should be conducted alone? You should express your opinion on this aspect.
Good suggestion, we have modified the discussion:
“The OGM technique involves the bulk analysis of the somatic variants and parsing out sub-clonal information may be more difficult to ascertain compared with KT. However, OGM not only has very high sensitivity as compared to KT but will also provide the variant allele fraction (VAF) of each variant to better understand the various abnormalities in these somatic samples. Although KT is a method able to look at individual clones, it fails to reveal cryptic translocation and is unbale to ascertain the identity of marker chromosomes, additions on terminal arms of chromosomes, or define the breakpoints involving oncogenes in complex abnormalities. OGM is unable to detect balanced centromeric translocations, however, these are not recommended for assessment by professional societies such as WHO or NCCN in hematological malignancies. Our findings are in line the previous publications recommending OGM as first tier test for heme malignancies. In cases of complex genome detected by OGM, KT might serve as a potential reflex test to assess the subclonal variants that are disease causing for a more comprehensive assessment of the cancer genome.”
- Figure 5: resolution must be improved, please.
Thank you for this comment, the resolution will be improved in the proof process.
- Figure S1: it should be moved to the manuscript as it describes the workflow of variants’ prioritization as disease-specific or pan-cancer specific. It should be improved to facilitate reader’s comprehension (for example the background should be white).
We appreciate this suggestion, and agree moving the workflow figure to the main text is an improvement. We have made revisions for clarity, and moved it to Figure 1 in the main text.
Figure 1. Overview of analysis and interpretation workflow. The curation and classification process consists of four steps: 1) disease- specific: Several organizations such as the World Health Organization (WHO), the National Health Service (NHS) and the National Comprehensive Cancer Network (NCCN) have published guidelines for assessing clinically relevant genomic regions for hematological malignancies. The analysts apply a filter to select for variants overlapping known loci associated with the disease of the sample (e.g. AML), and then classify the selected SVs as Tier 1A or 3; 2) Pan-hematological cancers: The analyst removes the first filter, and applies another filter to select for SVs seen in hematological cancers and classifies them as Tier 2 or 3; 3) Pan-cancer: SVs overlapping other cancer-associated genes are selected and classified as Tier 1B, 2, 3 or 4; and 4) Remaining variants: All remaining SVs that were fusions, or were >200 kbp are classified as Tier 1B, 2, 3 or 4.;
- Lines 159-160: “SVs were classified using a tiering system based on the ACMG guidelines adopted for OGM.” You should better specify how you generated the lists of variants (you cited also WHO, CNNC and NHS). Indeed, In the text you should better explain variants’ classification (Tier 1A, 1B, 2, 3, 4). In the Supplementary material you should list all the variants included in each classification (with the exception of Tier 4).
We thank the reviewer, this is a useful clarification. We have added text and a reference to clarify:
“Briefly, the four-tiered evidence-based SV classification followed professional recommendations for interpretation, variant classification, and reporting of genomic findings in neoplastic disorders [Mikhail et al. 2019] that were in turn informed by guidelines resourced form World Health Organization (WHO), National Comprehensive Cancer Network (NCCN), National Health Services (NHS, UK).”
We will also add a table to the supplement with all classified variants.
Submission Date
27 October 2023
Date of this review
10 Nov 2023 13:08:08

Reviewer 2 Report
Comments and Suggestions for Authors
The authors comprehensively present the value of OGM on the background of genetic examinations on hematological malignancies. I think this review article presents the value of use OGM instead KT, FISH and CMS. The authors in discusssion (351 - 358) present as well limitation of the method. I wouldn't change anything.
Author Response
Reviewer 2
Open Review
The authors comprehensively present the value of OGM on the background of genetic examinations on hematological malignancies. I think this review article presents the value of use OGM instead KT, FISH and CMS. The authors in discusssion (351 - 358) present as well limitation of the method. I wouldn't change anything.
Thank you to the reviewer for you for taking the time to review this manuscript and provide their kind thoughts about the value and quality of the manuscript.
Submission Date
27 October 2023
Date of this review
08 Nov 2023 13:34:14